# Structure, mechanism, and evolution of the last step in vitamin C biosynthesis

Alessandro Boverio[1,2], Neelam Jamil [2], Barbara Mannucci [3], Maria Laura Mascotti [1,4,5] ✉, Marco W. Fraaije [1] ✉ & Andrea Mattevi [2] ✉

Photosynthetic organisms, fungi, and animals comprise distinct pathways for vitamin C biosynthesis. Besides this diversity, the final biosynthetic step consistently involves an oxidation reaction carried out by the aldonolactone oxidoreductases. Here, we study the origin and evolution of the diversified activities and substrate preferences featured by these flavoenzymes using molecular phylogeny, kinetics, mutagenesis, and crystallographic experiments. We find clear evidence that they share a common ancestor. A flavin-interacting amino acid modulates the reactivity with the electron acceptors, including oxygen, and determines whether an enzyme functions as an oxidase or a dehydrogenase. We show that a few side chains in the catalytic cavity impart the reaction stereoselectivity. Ancestral sequence reconstruction outlines how these critical positions were affixed to specific amino acids along the evolution of the major eukaryotic clades. During Eukarya evolution, the aldonolactone oxidoreductases adapted to the varying metabolic demands while retaining their overarching vitamin C-generating function.

L-ascorbic acid (vitamin C) plays a pivotal role in nature. It is a highly soluble, carbohydrate-like compound that acts as a powerful anti-oxidant and co-substrate for many redox enzymes involved in a range of processes from biosynthesis of collagen, carnitine, and neuro-transmitters to chromatin modification[1–5]. While the ability of prokaryotes to produce vitamin C remains unclear, the majority of eukaryotes are endowed with autonomous vitamin C biosynthesis, typically using a sugar-phosphate as the biosynthetic precursor[6–11]. Specifically, in animals, vitamin C is synthesized in a pathway that converts glucose-6-phosphate into L-gulono-1,4-lactone, which is finally oxidized to L-ascorbic acid by L-gulono-1,4-lactone oxidase (GULO) [EC 1.1.3.8] (Fig. 1a)[12]. Fungi produce an ascorbate analog, D-erythroascorbic acid, through the oxidation of D-arabinono-1,4-lactone carried out by D-arabinono-1,4-lactone oxidase (ALO) [EC 1.1.3.37][13,14]. In photosynthetic organisms, L-ascorbic acid is produced in the Smirnoff-Wheeler pathway by the multi-step conversion of D-mannose-6-phosphate to L-galactono-1,4-lactone, the substrate for the final biosynthetic enzyme, L-galactono-1,4-lactone dehydrogenase (GalDH) [EC 1.3.2.3][9,15]. GULO, ALO, and GalDH are generally named aldonolactone oxidoreductases and form a group of sequence-related flavoenzymes belonging to the vanillyl-alcohol oxidase family[16–19]. They oxidize the C2 carbon atom of their monosaccharide substrates and thereby generate the characteristic, redox-reactive C2-C3 double bond of L-ascorbic acid and its analogs (Fig. 1a). Intriguingly, primates lost the ability to synthesize L-ascorbic acid due to extensive mutations in their GULO-encoding gene[20]. The lack of a functional GULO makes vitamin C an essential nutrient for humans, widely perceived as an iconic element of healthy diets[21].

The attainment of identical L-ascorbate or D-erythroascorbate products from various precursor molecules is facilitated by key differentiating factors among animal, fungal, and plant aldonolactone oxidoreductases. These enzymes diverge for the stereochemistry of

[1]Molecular Enzymology group, University of Groningen, Nijenborgh 4, 9747AG Groningen, The Netherlands. [2]Department of Biology and Biotechnology, University of Pavia, via Ferrata 9, 27100 Pavia, Italy. [3]Centro Grandi Strumenti, University of Pavia, Via Bassi 21, 27100 Pavia, Italy. [4]IMIBIO-SL CONICET, Facultad de Química Bioquímica y Farmacia, Universidad Nacional de San Luis, San Luis, Argentina. [5]Present address: Instituto de Histología y Embriología de Mendoza (IHEM)-CONICET-Universidad Nacional de Cuyo, 5500 Mendoza, Argentina. ✉e-mail: mlmascotti@mendoza-conicet.gob.ar; m.w.fraaije@rug.nl; andrea.mattevi@unipv.it

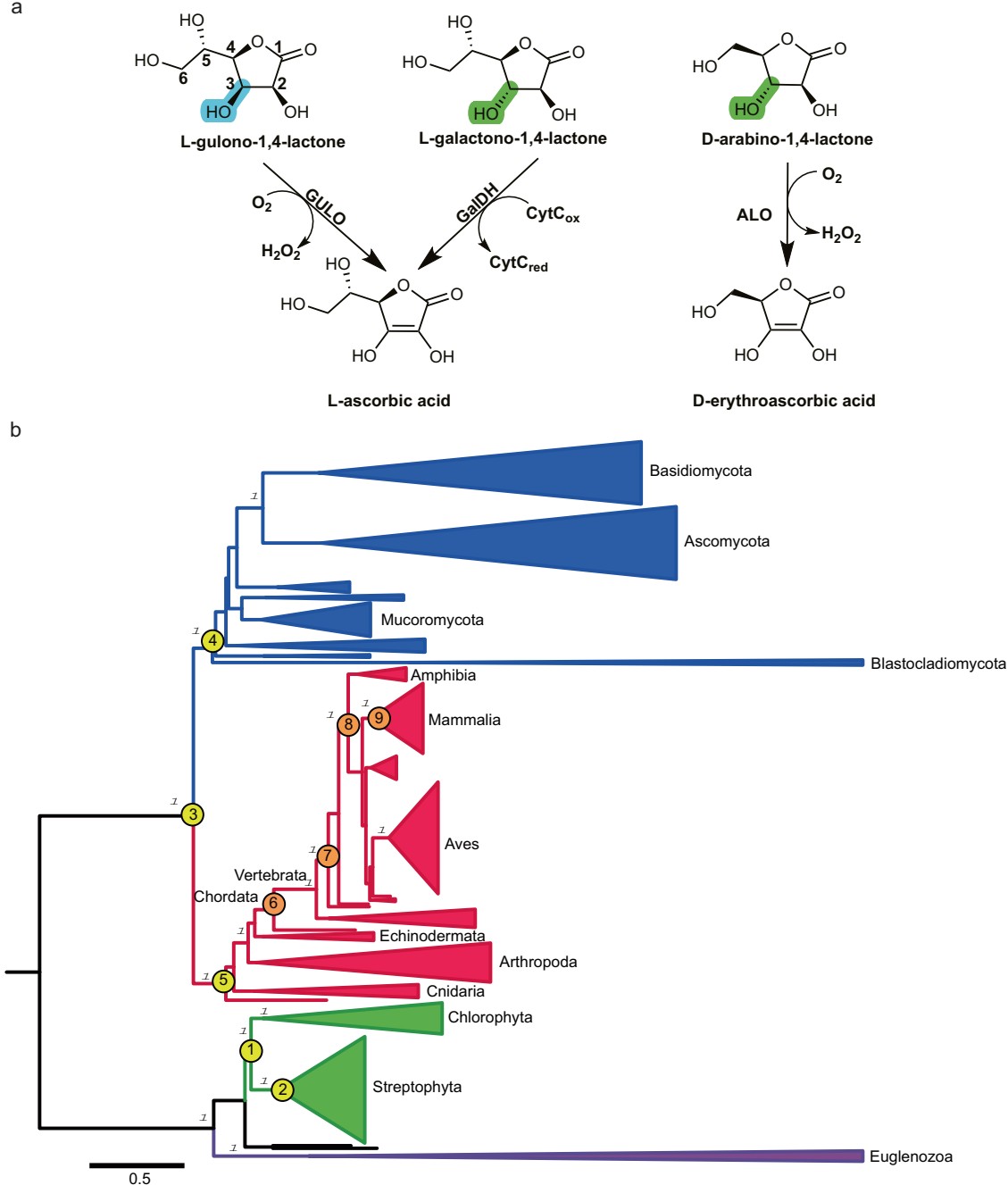

**Fig. 1 | Biochemistry and phylogeny of the aldonolactone oxidoreductase family. a** The reactions carried out by GalDH, GULO, and ALO. The chiral C3 carbon is highlighted. The numbering of the L-gulono-1,4-lactone atoms are shown for reference. Two cytochromes c (CytC) are reduced per turnover. **b** Condensed phylogenetic tree of the eukaryotic aldonolactone oxidoreductases where the taxonomy is indicated by the color of the branches: euglenozoa (purple), plants (green), metazoan (red) and fungi (blue). The major groups are labeled as reference. The targeted ancestors of: Viridiplantae (node 1, vGalDH), Streptophyta (node 2, sGalDH), Opisthokonta (node 3), Fungi (node 4) and Metazoa (node 5) are shown in yellow circles. The targeted internal ancestors: chordata (node 6, cGULO), jawed vertebrates (node 7, jGULO), tetrapods (node 8, tGULO) and mammals (node 9, mGULO) are shown in orange circles. The transfer bootstrap expectation values for the major divergences are given at the nodes in italics. The scale bar indicates the substitutions per site. For a fully annotated phylogeny please refer to Supplementary Fig. 1. With the only exceptions of the Opisthokonta (node 3), Fungi (node 4) and Metazoa (node 5), all ancestral proteins were characterized experimentally by this work.

their sugar substrates: the C3 carbon atom of the GULO and ALO substrates has the $R$ configuration, whereas GalDH operates on the $S$ enantiomer (Fig. 1a). They further differ in the bulkiness of the side chain on the substrate C5: a dihydroxyethyl moiety for GalDH and GULO and a smaller hydroxymethyl group for ALO. Another critical distinction stems from their reactivities with oxygen and preference for the co-substrate needed to regenerate the oxidized flavin. GalDH is a dehydrogenase that reacts poorly with oxygen. It is suggested that GalDH utilizes cytochrome c as its natural oxidant substrate, due to its close physical proximity to the respiratory complexes[22]. Conversely, GULO and ALO efficiently react with oxygen and operate as $H_2O_2$-producing oxidases. A final noticeable variation is the type of flavin incorporation. GULO and ALO bind the FAD covalently via an $8\alpha$-N1-histidyl linkage whereas GalDH binds it noncovalently.

Here, we investigate the functional divergence among aldonolactone oxidoreductases. We inferred several ancestors of plant GalDHs and animal GULOs that were expressed as recombinant proteins to delve into their biochemical properties, and solved the GalDH crystal structures with and without substrate bound[23–25]. Aided by the experimental analysis, we also reconstructed the sequences of the fungi and opisthokonta ancestors and predicted their functionality back to the divergence of these clades. Our findings reveal a common ancestor for all vitamin C-producing enzymes. Specific mutations in fungi and animals have facilitated the utilization of oxygen as a co-substrate, distinguishing them from plants in their evolutionary paths.

## Results

### Reconstruction of ancestral aldonolactone oxidoreductase sequences

The phylogeny of eukaryotic aldonolactone oxidoreductases was inferred by Maximum Likelihood method, employing a representative dataset including 268 sequences (Fig. 1b and Supplementary Fig. 1). The common ancestors of each major clade - viridiplantae, fungi and metazoa - were selected for experimental characterization. The common ancestors of land plants and algae (Viridiplantae, vGalDH) and plants (Streptophyta, sGalDH) were reconstructed with overall posterior probabilities ($\overline{PP}$) of 0.81 and 0.93, respectively (Supplementary Fig. 2)[26]. Comparison with the GalDH from *Arabidopsis thaliana* outlined 73% and 63% sequence identities with sGalDH and vGalDH, respectively. We noticed that most of these changes are localized in the N-terminal residues, likely a mitochondrial targeting sequence with a putative cleavage site as identified in several GalDHs from different species[15,27,28]. Therefore, we designed N-terminally truncated versions of sGalDH (Δ120, $\overline{PP}$ = 0.96) and vGalDH (Δ119, $\overline{PP}$ = 0.85) for the experimental characterization. The common ancestors of GULOs from chordata (cGULO), tetrapods (tGULO), jawed vertebrates (jGULO), and mammals (mGULO) were reconstructed with posterior probabilities greater than 0.90 (Supplementary Fig. 2). They showed 65-90% sequence identity with the extant GULO from *Mus musculus*. Finally, the common ancestor of the fungal D-arabino-1,4-lactone oxidases was

inferred ($\overline{PP}$ = 0.88) displaying a sequence identity of 40% with the extant ALO from *Candida albicans*. However, this ancestor was not expressed in soluble form, impairing its experimental characterization. More generally, the phylogenetic analysis suggests that the aldonolactone oxidoreductases follow a vertical inheritance pattern, as the topology of the protein tree is coincident with the one of the species tree.

### Ancestral GalDHs have conserved dehydrogenase activity

Both ancestral sGalDH and vGalDH were successfully expressed and purified. They displayed the characteristic properties of flavin-containing enzymes combined with remarkably high thermal stabilities as documented by melting temperatures of 69 °C and 87 °C for sGalDH and vGalDH, respectively (Supplementary Fig. 3, 4 and Supplementary Table 1). Extant *A. thaliana* GalDH was reported to act as dehydrogenase and, therefore, the two enzymes were initially analyzed by following the reduction of equine heart cytochrome *c* at 550 nm[15]. We noticed that their activities were strictly dependent on the ionic strength of the buffer, with a 20% activity loss using just 5 mM NaCl. pH optimum analyses further revealed that the GalDHs perform better under slightly alkaline conditions (Supplementary Fig. 5). Therefore, all subsequent experiments were conducted at pH 8, avoiding the usage of salts.

Both ancestors displayed a typical Michaelis-Menten behavior towards the physiological substrate, L-galactono-1,4-lactone, with relatively high $k_{cat}$ values of 8.7 ± 0.4 s$^{-1}$ for sGalDH and 18.6 ± 0.7 s$^{-1}$ for vGalDH. Noticeably, a similar activity was detected for L-gulono-1,4-lactone, the substrate of GULOs in animals' L-ascorbate biosynthesis (Fig. 1a). However, the K$_M$ values for L-gulono-1,4-lactone were ~125–250 times higher than those for L-galactono-1,4-lactone (Table 1 and Supplementary Table 2). Robust dehydrogenase activity was also detected using D-arabino-1,4-lactone, the five-carbon substrate of ALO, featuring the same stereochemistry as the GalDH substrate. In this case, the $k_{cat}$/K$_M$ values indicated a better specificity compared to L-gulono-1,4-lactone, yet 20- to 70-fold lower than the values measured for the physiological L-galactono-1,4-lactone (Table 1 and

**Table 1 | Steady-state kinetic parameters of ancient GalDHs[a,c] and GULOs[b,c]**

| | Substrate | K$_M$ (mM) | $k_{cat}$ (s$^{-1}$) | $k_{cat}$/K$_M$ (M$^{-1}$ s$^{-1}$) |
|---|---|---|---|---|
| **vGalDH[a]** | L-galactono-1,4-lactone | 0.0166 ± 0.0026 | 18.6 ± 0.7 | 1.1 × 10$^6$ |
| | L-gulono-1,4-lactone | 2.1 ± 0.3 | 8.3 ± 0.5 | 3.9 × 10$^3$ |
| | D-arabino-1,4-lactone | 1.1 ± 0.1 | 16.9 ± 0.8 | 1.5 × 10$^4$ |
| **vGalDH A113G(oxidase)[b]** | L-galactono-1,4-lactone | 0.040 ± 0.0030 | 9.8 ± 0.7 | 2.5 × 10$^5$ |
| | L-gulono-1,4-lactone | 10.4 ± 0.3 | 6.2 ± 0.7 | 6.0 × 10$^2$ |
| | D-arabino-1,4-lactone | 0.13 ± 0.002 | 3.3 ± 0.2 | 2.5 × 10$^4$ |
| **vGalDH A113G(dehydrogenase)[a]** | L-galactono-1,4-lactone | 0.053 ± 0.012 | 17.1 ± 1.5 | 3.2 × 10$^5$ |
| | L-gulono–1,4-lactone | 4.2 ± 0.6 | 7.2 ± 0.4 | 1.7 × 10$^3$ |
| | D-arabino–1,4-lactone | 1.3 ± 0.2 | 10.5 ± 0.7 | 8 × 10$^3$ |
| **vGalDH G413N[a]** | L-galactono-1,4-lactone | 0.015 ± 0.002 | 7.0 ± 0.2 | 4.6 × 10$^5$ |
| | L-gulono-1,4-lactone | 0.284 ± 0.034 | 2.1 ± 0.1 | 7.4 × 10$^3$ |
| | D-arabino-1,4-lactone | 6.3 ± 0.9 | 5.7 ± 0.3 | 9 × 10$^2$ |
| **cGULO[b]** | L-galactono-1,4-lactone | 11.8 ± 1.2 | 3.5 ± 0.3 | 3.0 × 10$^2$ |
| | L-gulono-1,4-lactone | 6.1 ± 0.7 | 3.2 ± 0.1 | 5.2 × 10$^2$ |
| | D-arabino-1,4-lactone | 48.6 ± 24 | 0.6 ± 0.1 | 1.23 × 10$^1$ |
| **mGULO[b]** | L-galactono-1,4-lactone | 1.9 ± 0.2 | 4.2 ± 0.1 | 2.2 × 10$^3$ |
| | L-gulono–1,4-lactone | 0.8 ± 0.1 | 6.1 ± 0.1 | 7.6 × 10$^3$ |
| | D-arabino-1,4-lactone | 3.3 ± 0.4 | 0.5 ± 0.1 | 1.5 × 10$^2$ |

[a]Measured at 25 °C in 50 mM potassium phosphate buffer pH 8 with 50 µM cyt c. Activities below 0.2 s$^{-1}$ were measured when oxygen was used in place of cytochrome c. Data for sGalDH are listed in Supplementary Table 2.
[b]Measured at 25 °C in 50 mM potassium phosphate buffer pH 8 by monitoring oxygen consumption. Data for jGULO and tGULO are listed in Supplementary Table 2.
[c]Supplementary Figs. 6–17 and Source Data show the kinetic data.

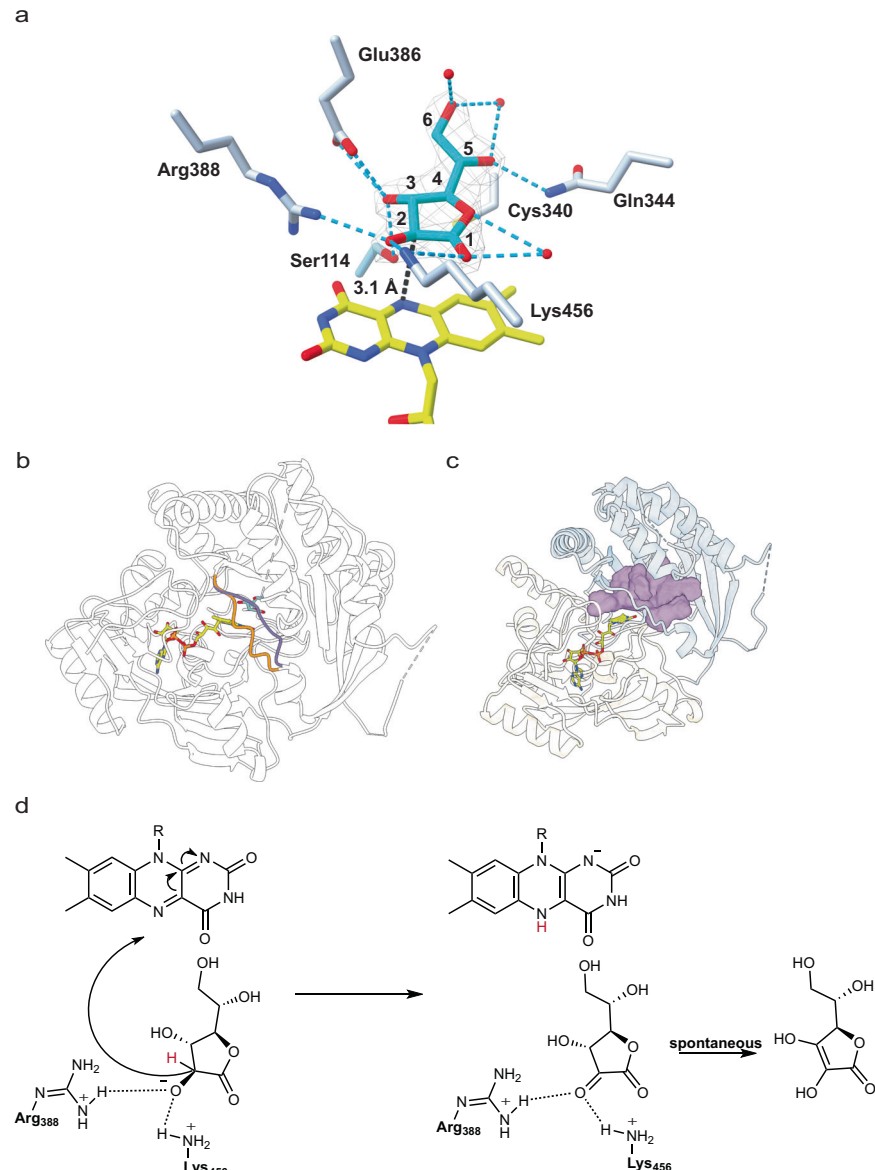

**Fig. 2 | The crystal structure of vGalDH. a** Closed-up view of substrate binding. L-galactono-1,4-lactone is shown with cyan carbons together with the polder omit map, calculated by excluding the ligand and countered at 3.5 σ[48]. Hydrogen bonds are represented in light blue. **b** A gating loop (residues 337-344) moves upon substrate binding. The conformation of the unligated structure is shown in orange whereas the substrate-bound conformation is in purple. The flavin and substrate carbons are yellow and cyan, respectively. **c** Closure of the gating loop constricts the access to the active site and creates the catalytic cavity in front of the flavin. The

surface of the cleft at the domain interface is shown in purple. **d** Proposed mechanism for substrate oxidation in the aldonolactone oxidoreductases with reference to vGalDH. The substrate 2-alkoxide anion is stabilized by the interaction with the Arg388·Lys456 pair. The C2 carbon transfers a hydride anion to the flavin, producing the 1,2-diketo form of ascorbic acid. The reduced flavin will be re-oxidized by oxygen in GULO and ALO or cytochrome c in GalDH (Fig. 1a). Arg388 and Lys456 are conserved among aldonolactone oxidoreductases (Supplementary Table 4).

Supplementary Table 2). For all three substrates, we observed only marginal oxidase activities (< 0.2 s$^{-1}$). Collectively, these data demonstrate that early plant GalDHs acted as dehydrogenases featuring a marked preference for L-galactono-1,4-lactone and its C3 configuration.

**Substrate recognition and structural basis of catalysis in GalDH**
To elucidate the structural features of GalDH and how it interacts with its substrate, we determined the crystal structures of the viridiplantae ancestor without (1.9 Å resolution) and with L-galactono-1,4-lactone (2.1 Å resolution; Supplementary Table 3). As a member of the vanillyl-alcohol oxidase family[18,29], the structure of vGalDH can be divided into flavin- (residues 1-189, 452-511) and substrate-binding (190-451)

domains (Supplementary Fig. 18). The FAD is embedded in the flavin domain with the isoalloxazine ring in the classical position at the domain interface where an open cleft forms the active site on the *si* side of the flavin. Soaking in a substrate-containing solution led to the bleaching of the crystals and the resulting electron density clearly indicated the presence of the bound substrate. We, therefore, conclude that the crystals accumulate the substrate-bound reduced state, outlining the structure of the Michaelis complex (Fig. 2a). Upon substrate binding, the movement of a gating loop (residues 337-344) constricts the access to the active site, creating the catalytic cavity (Fig. 2b, c). L-galactono-1,4-lactone adopts a pro-catalytic conformation with its C2 carbon at 3.1 Å distance from the isoalloxazine ring and C2 hydrogen predicted to point towards the flavin N5 position. The

surrounding side chains establish a hydrogen bond network with the ligand hydroxyl groups. These interactions are enforced by the closure of the gating loop with two of its residues, Cys340 and Gln344, moving closer to the sugar to interact with its 1 and 5 oxygens. Both residues are conserved in the two ancestors and the extant *A. thaliana* GalDH, underlying their role in substrate recognition and catalysis (Supplementary Table 4)[30]. Arg388 and Lys456 critically position their positively charged side chains in contact with the substrate O2 atom, confirming previous hypotheses about the importance of Arg388 for enzyme activity[31]. In particular, these residues may promote the binding of the alkoxide anion of the substrate, which in turn facilitates hydride transfer from the C2 position to the flavin N5, leading to substrate oxidation (Fig. 2d). The resulting 1,2-diketo product will then spontaneously tautomerize to the more stable enediol form of vitamin C[32].

## L-Gulono-1,4-lactone oxidases have less stringent substrate preferences

We next investigated the animal branch of aldonolactone oxidoreductase evolution by resurrecting GULOs from chordata, tetrapods, jawed vertebrates, and mammals (Fig. 1b). The *Escherichia coli*-expressed recombinant enzymes needed detergents for extraction and purification implying that they are membrane-associated. Sequence analyses did not clearly indicate which residues are involved in the membrane association. Nevertheless, AlphaFold predicts a low confidence region (residue 241 to 256) that might be involved in membrane interaction. SDS-PAGE and subsequent in-gel flavin fluorescence analysis confirmed that, differently from GalDHs, the FAD is covalently bound in all obtained proteins (Supplementary Fig. 4B, C). All four proteins displayed pronounced thermal stabilities with melting temperatures ranging from 66 °C for mGULO to 80 °C for cGULO (Supplementary Table 1). The activities of ancient GULOs towards the physiological substrate, L-gulono-1,4-lactone were investigated. Monitoring oxygen consumption, $k_{cat}$ values in the range 1-8 s$^{-1}$ were found for all ancestral enzymes whereas $K_M$ values in the sub-millimolar range were measured for mGULO and tGULO (Table 1 and Supplementary Table 2). We further observed that also L-galactono-1,4-lactone (*i.e.*: GalDH substrate; Fig. 1A) and D-arabino-1,4-lactone (ALO substrate) are effectively oxidized with catalytic efficiencies comparable to those measured for L-gulono-1,4-lactone. We conclude that GULOs feature only a minimal preference for L-gulono-1,4-lactone over L-galactono-1,4-lactone as opposed to pronounced selectivity exhibited by GalDHs.

To delve into the mechanism, a pre-steady state kinetics analysis was conducted. mGULO was analyzed using the stopped flow to determine the rates of the reductive and oxidative half-reactions (Supplementary Fig. 19). The reductive half-reaction was measured in an anaerobic environment at different substrate concentrations. The rates of reduction were sustained with $k_{red}$ values of 89.8 ± 2.5 s$^{-1}$ for L-gulono-1,4-lactone and 124.5 ± 3.1 s$^{-1}$ for L-galactono-1,4-lactone. On the other hand, D-arabino-1,4-lactone resulted in a slower reduction rate, 14.2 ± 2.3 s$^{-1}$, outlining the importance of the terminal C6 carbon for optimal substrate accommodation in the active site (Fig. 1a). The dissociation constants were in the low millimolar range with the highest affinity for L-gulono-1,4-lactone ($K_d$ = 1.8 ± 0.2 mM). L-galactono-1,4-lactone showed a slightly lower affinity ($K_d$ = 4.1 ± 0.4 mM) confirming a mild binding preference for L-gulono-1,4-lactone over L-galactono-1,4-lactone. Finally, the oxidative half-reaction was also analyzed. The anaerobically reduced enzyme was mixed with different oxygen concentrations showing a re-oxidation rate ($k_{ox}$ = 6.4 s$^{-1}$ ± 0.3 s$^{-1}$ at atmospheric oxygen concentration) that matches the $k_{cat}$ (Table 1). Based on these findings, we conclude that GULOs retained their oxidase activity along the evolution of chordata with the flavin re-oxidation being the rate-limiting step in catalysis.

## GalDH and GULO produce L-ascorbic acid during catalysis

In nature, L-ascorbic acid can exist in three different redox states: the fully reduced form (L-ascorbic acid), the mono-oxidized form (semidehydroascorbate), and the fully oxidized state (dehydroascorbate). L-ascorbate is the most abundant and useful form acting in biological reactions[33]. To confirm that the resurrected plant GalDH and mammal GULO ancestors can produce vitamin C and address its redox state, reaction products were analyzed with ultra-high-pressure liquid chromatography coupled with high-resolution mass spectrometry (Supplementary Fig. 20). Reactions were performed using L-gulono-1,4-lactone and L-galactono-1,4-lactone as substrates. In all the cases, the major product found was L-ascorbic acid. A minor part of dehydroascorbate was also observed, while no traces of semidehydroascorbate were found. Hence, aldonolactone oxidoreductases actively generate the fully reduced L-ascorbate as the major reaction product.

## On the origin of the oxidase *vs.* dehydrogenase distinction

The biochemical elucidation of ancestral GalDHs and GULOs clearly demonstrates that their respective dehydrogenase-like (*i.e.*: poor), and oxidase-like (sustained) reactivities with oxygen are ancient and innate features of the two enzymes. Comparison of their active sites highlighted a remarkably high degree of conservation, narrowing down the differences in substrate specificity and electron acceptor usage to a few key amino acids. Multiple sequence alignments and structural analysis outlined a noticeable discrepancy at position 113 (vGalDH numbering). GalDH displays a strictly conserved alanine, whereas fungal ALO and metazoan GULO possess a glycine at this position (Fig. 3a–d, Supplementary Table 4 and Supplementary Fig. 21). With the GalDH crystal structure at hand, we noticed that Ala113 is in direct contact with the N5-C4a locus of the flavin at the edge of the cofactor ring (Fig. 3a). Previous studies have shown that the mutation A113G increases the oxidase activity in an extant plant GalDH[34]. Expanding on this background knowledge, we asked whether the proposed gatekeeper and oxygen-controlling function of Ala113 is a conserved feature of GalDH clade. The A113G vGalDH and sGalDH mutants were therefore prepared and characterized. Steady-state kinetics analysis revealed an evident increase (from 15 to 200-fold) of their $k_{cat}$ values for all three analyzed sugar substrates using oxygen as co-substrate (Table 1 and Supplementary Table 2). In fact, the oxidase activities of the A113G enzymes match the values measured for GULOs. Stopped-flow experiments further confirmed that the A113G mutation largely improves the reactivity with oxygen; 300-fold for vGalDH ($k_{ox}$ = 67.5 ± 5.3 s$^{-1}$ at atmospheric oxygen concentration) and 70-fold for sGalDH ($k_{ox}$ = 14.7 ± 2.4 s$^{-1}$) (Fig. 3e, f). To determine the structural basis behind this feature, we set out to solve the three-dimensional crystal structure of the A113G vGalDH (Supplementary Table. 3). The removal of the Ala113 methyl side chain creates an access route for the oxygen to diffuse to the flavin N5-C4a locus. This feature is manifested by the presence of an ordered water molecule on the *re* side of the flavin, at 3.8 Å from the C4a (note that the substrate site is on the flavin *si* side, Fig. 3g). In the wild-type structure, a water molecule in this position would be prohibited owing to the overlap with the Ala113, consistent with the blockade imposed by its side chain. In addition to the N5-interacting residue at position 113, more residues in the flavin environment likely contribute to the oxygen reactivity in GULO and its repression in GalDH. Nevertheless, even a simple, single-site mutation can drastically perturb this enzymatic property.

## A single residue is behind the substrate specificity of GalDH

In addition to the different oxygen reactivities, the key functional distinction between GalDH and GULO concerns the preference for the reductant substrates. GalDH is up to 1000-fold more efficient on L-galactono-1,4-lactone than L-gulono-1,4-lactone, whereas GULO

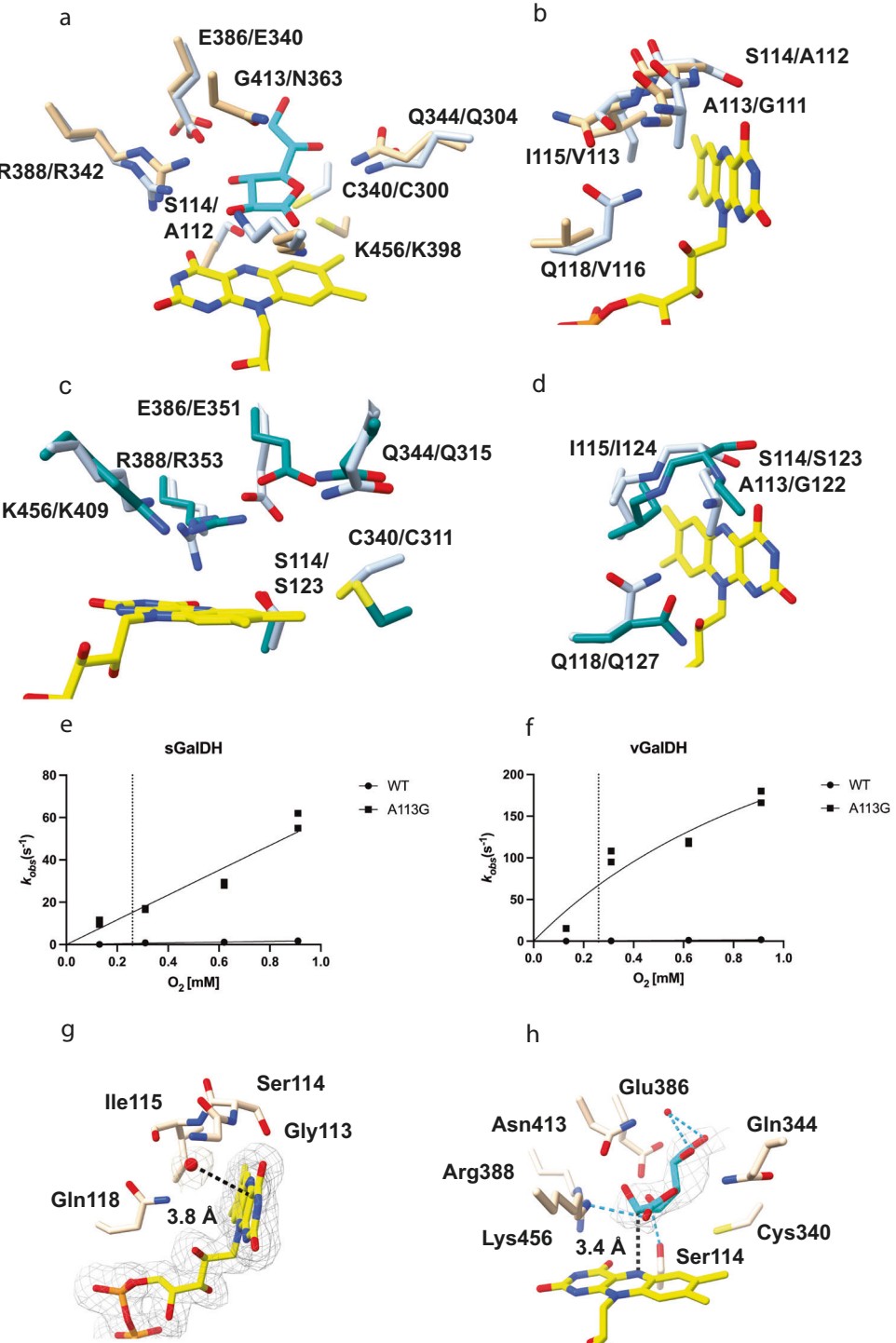

**Fig. 3 | vGalDH variants. a-b** Superposition between the substrate- and flavin-binding residues of the vGalDH crystal structure (light steel blue) and Alpha Fold–predicted mGULO (light brown)[49,50]. The flavin and L-galactone-1,4-lactone carbons are in yellow and cyan. Residues are labeled with the format vGLADH/mGULO. **c-d** Superposition between the substrate- and flavin-binding residues of the vGalDH crystal structure (light steel blue) and AlphaFold–predicted fungal ancestor's ALO (dark cyan)[49,50]. The flavin and L-galactone-1,4-lactone carbons are in yellow and cyan. Residues are labeled with the format vGALDH/ALO. **e–f** The re-oxidation rates of sGalDHs and vGalDH proteins were measured using stopped-flow methods by following the absorbance of the oxidized flavin at 450 nm; n = 2 independent experiments, individually plotted as dots. The dotted lines correspond to the atmospheric concentration of dioxygen (0.26 mM). The experiments were performed with 5–10 μM enzyme concentrations (after mixing). Source data are provided. **g** Closed-up view of the flavin *re* side in the crystal structure of vGalDH A113G mutant. Water molecules are in red. The polder omit map was calculated by excluding the FAD and nearby water molecule and countered at 3.5 σ. **h** Crystal structure of vGalDH G413N variant in complex with L-gulono-1,4-lactone. The flavin carbons are in yellow and L-gulono-1,4-lactone carbons are in cyan. The polder omit map was calculated by excluding the ligand and countered at 3.5 σ. Hydrogen bonds are represented in light blue.

features a less stringent specificity with only a marginal inclination (up to 10-fold) towards L-gulono-1,4-lactone (Table 1). The structural comparison shows that the only conspicuous change between the substrate-binding sites of the two enzymes is the replacement of Gly413 in GalDH to Asn363 in GULO (residue numbers refer to vGalDH and mGULO; Supplementary Table 4, Fig. 3a and Supplementary Fig. 21). We sought to unveil the role of this amino acid exchange by introducing the G413N mutation in wild-type and A113G GalDHs. The A113G-G413N proteins were poorly expressed and featured lower activities. Encouragingly, however, we found that the single G413N mutants had sustained activities. Both G413N sGalDH and vGalDH displayed a selective decrease in the $K_M$ for L-gulono-1,4-lactone and no drastic alterations in the $K_M$ for L-galactono-1,4-lactone or the respective $k_{cat}$ values (Table 1 and Supplementary Table 2). We further performed the crystallographic analysis of G413N vGalDH at 2.3 Å resolution using crystals that were soaked in a solution comprising L-gulono-1,4-lactone. The B-factors of the bound sugar were notably high (Supplementary Table 3). However, despite this, the electron density was of sufficient quality to conclude that the conformation and orientation of L-gulono-1,4-lactone are highly similar to that of L-galactono-1,4-lactone bound to the wild-type enzyme, with the obvious exception of the 3-hydroxyl group position (Fig. 1a, 3h). The substrate-binding environment is likewise mostly retained in the mutant structure. The only differences affect Glu386, which shifts towards Arg219 and no longer forms an H-bond with the ligand, and the newly formed H-bond between the ligand O3 and Arg388 (Fig. 3h). The carbamide group of the introduced Asn413 is nestled on the sugar ring and in van der Waals contact with the substrate O3. Collectively, these data demonstrate that the GalDH *versus* GULO distinction can be primarily explained by very few amino acid changes in an otherwise remarkably well-conserved catalytic cavity.

## Discussion

By using ancestral sequence reconstruction, we sought insights to decipher the biochemical features of the aldolonolactone oxidoreductases at the emergence of the major clades in Eukarya evolution. The dehydrogenase lineage features an alanine residue at the site in direct contact with the flavin N5-C4a locus (Fig. 3b). This amino acid is replaced by an oxidase-enabling glycine in the progenitor of fungi and metazoa (opisthokonta ancestor, node 3 in Fig. 1b). This pivotal mutation is accompanied by the installation of the covalent flavin-protein linkage by replacing a Leu with a His side chain in proximity to the flavin dimethylbenzene ring. Subsequent conservation of this flavin-interacting Gly/His residue pair accounts for why covalent flavinylation and oxidase activity continue to represent two traits that distinguish modern fungal and animal aldonolactone oxidoreductases from their plant counterparts (Supplementary Table 4).

The physiological substrates of GalDH and GULO, L-galactono-1,4-lactone and L-gulono-1,4-lactone, differ in the chirality of their C3 carbon (Fig. 1a). GalDHs exhibit a pronounced preference for L-galactono-1,4-lactone whereas GULOs feature a more relaxed profile as they can act with similar efficiencies on both sugars. A critical change in the composition of the substrate cavity made this functional property possible. Ancient GalDHs feature a substrate-interacting glycine that has been conserved throughout evolution of GalDH. This site was instead mutated in the metazoan ancestor (node 5 in Fig. 1b) introducing an asparagine side chain that enables the activity on both stereoisomers as demonstrated by our experimental analysis. This feature has been maintained in extant GULOs (Supplementary Table 4).

In conclusion, our study sheds light on the evolutionary trajectory and functional transitions experienced by the aldonolactone oxidoreductases responsible for ascorbic acid generation. These eukaryotic enzymes show an early divergence in two lineages, a dehydrogenase-like one in plants and an oxidase-like one in fungi and metazoa. A limited set of stepwise mutations along the respective evolutionary branches gave rise to the functional properties specifically characterizing fungal ALOs and animal GULOs. Hence, starting from an early aldonolactone oxidoreductase, Nature tuned the enzymatic functions to distinct metabolic pathways and electron acceptors without compromising on the aim: the biosynthesis of vitamin C.

## Methods

### Ancestral sequence reconstruction

To infer the aldonolactone oxidoreductases phylogeny, a dataset was constructed by performing BLASTp searches in GenBank (non-redundant protein sequences (nr) database) using GalDH sequence from *A. thaliana* [Uniprot: Q9SU56] and GULO sequence from *M. musculus* [Uniprot: P58710] as queries. Searches were conducted by classes or orders from the Eukarya domain and sequences from at least two species with fully annotated genomes were collected. The obtained phylogeny was rooted following the species tree, guided by TimeTree knowledge database[35]. A total of 268 sequences were collected (124 from metazoa, 81 from fungi, and 63 from viridiplantae and euglenozoa; Supplementary Data 1) and a multiple sequence alignment was created using MAFFTv7[36]. Single-sequence extensions and gaps were manually trimmed. Phylogeny was inferred using the maximum likelihood method in RAxMLv8.2.10[37] with 500 rapid bootstraps and automated model selection. Bootstrap values were subjected to transfer bootstrap expectation in BOOSTER[38].

Ancestral Sequence Reconstruction was performed using the codeml package in PAMLv4.9[39]. Sequences were analyzed using an empirical substitution matrix, an empirical equilibrium amino acid frequency (model= 3), and 4 gamma categories. The length of the ancestors was treated by parsimony analyzing the presence/absence of gaps in the targeted nodes. The posterior probability distribution was analyzed at each site for the selected nodes: sGalDH ($\overline{PP} = 0.93$), vGalDH ($\overline{PP} = 0.81$), mGULO ($\overline{PP} = 0.99$), tGULO ($\overline{PP} = 0.97$), jGULO ($\overline{PP} = 0.94$), cGULO ($\overline{PP} = 0.90$), metazoan ancestor GULO ($\overline{PP} = 0.89$), fungal ancestor ALO ($\overline{PP} = 0.88$), Opistokhonta ancestor ($\overline{PP} = 0.86$). The sites showing a $PP < 0.8$ and a second-best state with a $PP > 0.2$ were considered ambiguously reconstructed. By aligning the two full-length ancestral states with the extant GalDH from *A. thaliana*, a conserved mitochondrial target sequence with an XR/YA cleavage site was identified and it was used as trimming point to produce the N-truncated protein versions (Supplementary Fig. 2).

### Cloning, mutagenesis, and expression

Genes for all the selected nodes (Fig. 1b) were codon optimized for suitable expression in *E. coli* (TWIST bioscience) with BSAI sites at the 5' and 3' termini and cloned into a pBAD His-SUMO vector with the Golden Gate cloning methodology[40]. Cloning procedures were validated by sequencing. Mutants were prepared using the QuickChange protocol[41]. Primers were ordered form Microsinth. PCR reaction (25 µL) was composed as follow: 1 µL of primer FW (10 µM) and 1 µL of primer RV (10 µM), 100 ng of template, 0.4 µL of DMSO, PfuUltra II Hotstart PCR (12.5 µL) and MQ water up to 25 µL. Primers used for mutagenesis are available in the Supplementary Information section (Supplementary Table 5).

For transformation, 2 µL of plasmid was added to 100 µL of RbCl-competent *E. coli* NEB10®-β cells. After 30 min of ice incubation and 35 s of heat shock (42 °C), 250 µL of LB medium were added and the cells were recovered for 1 h at 37 °C. 50 µL of recovered cells were plated on LB-agar supplemented with ampicillin (100 µg/mL⁻¹) and incubated overnight at 37 °C. A single colony was picked and transferred to a 5 mL pre-inoculum of LB-amp (100 µg/mL⁻¹) grown overnight at 37 °C and subsequently used to inoculate a 2 L baffled flask containing 1 L of TB medium supplemented with ampicillin (100 µg/mL⁻¹). Flasks were incubated until an OD of 0.8-1 was reached. Protein expression was then induced with L-arabinose (0.02%), and cultures were left at 24 °C for 24 h. Cells were harvested by centrifugation

(6000 g, 15 min, 4 °C). sGalDH, vGalDH, and four GULOs (mGULO, tGULO, jGULO, and cGULO) were successfully expressed.

## Enzyme purification

The reagents used for protein expression, purification, and characterization were obtained from Merck (Darmstadt, Germany). GalDH-expressing cells were re-suspended in buffer A (100 mM potassium phosphate pH 8.0, 500 mM NaCl). The ratio of mass to volume was 1:4. Buffer A was supplemented with phenylmethylsulfonyl fluoride (0.10 mM) to prevent protein degradation, DNase I (5 µg g$^{-1}$ of cell), and FAD (0.1 mM). The re-suspended cells were disrupted either through high-pressure homogenizer or sonication. Cells were disrupted by passing three times through the high-pressure homogenizer. For sonication, a protocol of 5 s ON and 5 s OFF with an amplitude of 75% was used for a total of 10 min. After lysis, cells were centrifuged (56000 g for 1 h). The supernatant was then loaded on a gravity column containing 3 mL of Ni Sepharose Excel previously equilibrated with buffer A. Following a washing step (4 column volumes) with buffer B (50 mM potassium phosphate pH 8.0, 500 mM NaCl, 20 mM imidazole), the enzymes were eluted (2 columns volume) with buffer C (50 mM potassium phosphate pH 8.0, 500 mM NaCl, 500 mM imidazole). The buffer was then exchanged against 50 mM potassium phosphate pH 8.0. GalDH concentrations were determined by measuring the absorbance at 449 nm and using an extinction coefficient experimentally determined of $\varepsilon_{449} = 12.03$ mM$^{-1}$ cm$^{-1}$.

Ancestral GULOs were purified as membrane-associated enzymes. After resuspending the cells in buffer A, they were disrupted through a high-pressure homogenizer. After centrifugation (56000 g for 1 h), membranes were re-suspended in buffer A supplemented with 0.5% Triton X-100 reduced (TRX-100-R) and incubated overnight at 4 °C on a rotating wheel. The mixture was then centrifuged (56000 g for 1 h), and the obtained supernatant was loaded on a gravity column containing 5 mL of Ni Sepharose Excel resin previously equilibrated with buffer A supplemented with 0.05% TRX-100-R. The resin was then washed with buffer B supplemented with 0.05% TRX-100-R and eluted with buffer C supplemented with 0.05% TRX-100-R. The buffer was then exchanged using a centrifugal filter unit (50 kDa cut-off) and multiple washes with 50 mM potassium phosphate pH 8 plus 0.05% (v/v) TRX-100-R. GULOs concentrations were determined by measuring the absorbance at 448 nm and using an extinction coefficient of $\varepsilon_{448} = 12.5$ mM$^{-1}$ cm$^{-1}$. The enzyme purity was confirmed through SDS-page analysis.

## Enzyme activity

L-galactono-1,4-lactone, L-gulono-1,4-lactone, and D-arabino-1,4-lactone were obtained from Sigma-Aldrich. Dehydrogenase activities were evaluated by monitoring the reduction of 50 µM equine cytochrome c at 550 nm ($\varepsilon_{515} = 21$ mM$^{-1}$ cm$^{-1}$) at 25 °C using a Cary100 spectrophotometer (Agilent). Equine heart cytochrome c (Sigma-Aldrich) was solubilized in storage buffer. Oxidase activities were assayed using an Oxygraph Plus system (Hansatech Instruments Ltd.). Initial rates were determined by analyzing the initial part of the oxygen depletion curve. In all assays, the reactions were initiated by adding the enzymes (20 nM GalDH, 1.0 µM GULO) to the system. Data were processed in GraphPad Prism and fitted using a standard Michaelis Menten formula resulting in K$_M$ and $k_{cat}$ values. All experiments were performed in duplicates.

## Thermal stability

Enzyme thermal stability was addressed using the Tyco instrument from NanoTemper. Intrinsic fluorescence at 350 and 330 nm of tryptophan and tyrosine was measured by applying a consecutive temperature gradient from 35 °C to 95 °C. 1 mg/ml of protein was used. Experiments were carried out in duplicates.

## pH optimum and salt dependency

pH optimum was performed monitoring the cytochrome c (50 µM) reduction at 550 nm at 25 °C. 100 nM of purified enzyme solutions in 50 mM potassium phosphate pH 8 were used together with 500 µM of L-galactono-1,4-lactone. The total reaction volume was 100 µL. The reaction was initiated by adding the enzyme (20 nM GalDH, 1 µM GULO). Salt inhibition analysis was performed in the same above-mentioned condition varying the ionic strength of the buffer from 0 of NaCl to 300 mM of NaCl. Experiments were carried out in duplicates.

## Fast kinetics

Experiments were performed using a SX20 stopped-flow spectro-photometer equipped with a photomultiplier tube (PMT) and a photo diode array detector (PDA) (Applied Photophysics, Surrey, UK). Experiments were carried out with the enzymes fused to the His-SUMO tag at a concentration of 5-10 µM (enzyme concentration after mixing). All solutions were prepared using 50 mM potassium phosphate pH 8.0. Reactions were run in duplicates by mixing 50 µL of each solution at 25 °C. To follow the oxidative half-reaction, enzymes were made anaerobic by flushing the vial with nitrogen for 15 min. Moreover, 5 mM of glucose and 0.3 µM of glucose oxidase (*Aspergillus niger*, type VII, Sigma-Aldrich) were added to the solution to remove the leftover of oxygen. The flavin cofactor was subsequently reduced by adding sodium dithionite. The resulting mixture was then incubated at room temperature until the enzyme was completely reduced as indicated by the bleaching of the color. To evaluate the $k_{ox}$, the anaerobically reduced enzyme solutions were mixed with four different oxygen concentrations: air-saturated buffer; equal volumes of 100% argon buffer and 100% O$_2$ buffer; 100% O$_2$ buffer; 100% O$_2$ buffer on ice. The $k_{ox}$ values were obtained by fitting the transient kinetic traces obtained with the PMT detector with a first-order exponential function using the software linked to the stopped-flow instrument (Applied Photophysics, Surrey, UK). To determine the reductive half-reaction, the system was made anaerobic by flushing the two vials with nitrogen for 15 min. Glucose oxidase was also used to remove residual molecular oxygen. Different concentrations of substrates were used and the $k_{obs}$ values were obtained by fitting the transient kinetic traces obtained with the PMT detector with standard first- or second-order exponential decay functions using the software linked to the stopped-flow instrument (Applied Photophysics, Surrey, UK). The resulting dependence was fit to an equation that describes a rectangular hyperbola according to: $k_{obs} = [S]k_{red}/K_d + [S]$ where $k_{red}$ is the rate constant for flavin reduction and K$_d$ is the dissociation constant for the sugar substrate. The analyses of kinetic traces and the respective raw data are provided as Source Data.

## Product characterization

Reaction mixtures (1 mL) were prepared to analyze the L-ascorbic acid formation. The sample contained 1 µM of enzyme, 5 mM or 2.5 mM substrate (L-galactono-1,4-lactone or L-gulono-1,4-lactone), 50 µM of cytochrome c in 50 mM HEPES pH 8. Variants with an improved oxidative activity were tested without the usage of cytochrome c in the reaction. The reactions were set for 2 h at 25 °C with 180 rpm shaking. Control experiments were run with all assay components except the enzyme. To evaluate the vitamin C production 50 µL were taken and diluted with 60 µL 50 mM HEPES pH 8 and then quenched with the addition of 100 µL of 50% metaphosphoric acid and centrifuged for 10 min at 16,000 g. This procedure resulted in a 4.2-fold dilution. Samples were analyzed by electrospray ionization quadrupole time-of-flight high-resolution mass spectrometry, ESI-QTOF-HRMS, on an X500B QTOF system (SCIEX, Framingham, MA 01701 USA) equipped with the Twin Sprayer ESI probe coupled to an ExionLC system (SCIEX). The SCIEX OS software 3.0.0 was used as an operating platform. Chromatographic separation was carried out using an LC column ZORBAX Extended-CI8 (2.1 x 50 mm, 5 µm). The mobile phase

consisted of water (A) and acetonitrile (B) (both including 0.1% aqueous formic acid, v/v), and the flow rate was 0.3 ml/min. The liquid phase gradient was set as 0 to 1 min 0% B, 1 to 10 min 10% B, 10 to 12 min 40% B, 12 to 13 min 40% B, 13 to 15 min 100% B, 15 to 16 min 0% B, for four minutes. For MS detection the following parameters were applied: Curtain gas 30 psi, Ion source Gas 1 45 psi, Ion source Gas 2 55 psi, Temperature 450 °C. The full-scan range of m/z 20 to 500 was monitored in negative mode, with a Spray voltage of −4500 V, a declustering potential of −60 V, and a collision energy of −10 V. Mass calibration was performed with the ESI Negative Calibration solution suitable for the Sciex X500 system (SCIEX), that consists of a mix of known molecular weight chemicals.

## Crystal structure determination

After cleaving the SUMO tag, the purified enzyme was loaded on a Superdex 75 10/300 column (Cytiva) using an ÄKTA pure. Three wavelengths (280, 375, and 450 nm) were set for monitoring the protein run during the Size Exclusion Chromatography experiment. The central fractions of the protein peak were pooled together and concentrated until a final concentration of 17 mg/mL was obtained based on the flavin absorption peak. The final ratio A280/A450 was 7.4 indicating how the enzyme is fully present in the holo-form. Several crystallization conditions were tested using an Oryx8 robot (Douglas Instrument) at 20 °C. Elongated needle-like yellow crystals grew in several conditions, all containing PEG and lithium salts. The best diffracting crystals were obtained by mixing equal volumes of protein (17 mg vGalDH/mL in 50 mM HEPES pH 7.5) and reservoir (0.2 M LiNO$_3$ and 20% w/v PEG3350) solutions. A solution containing the mother liquor 25% v/v glycerol and 80 mM of the substrate of choice (L-galactono-1,4-lactone or L-gulono-1,4-lactone) was used as cryoprotectant. Crystals were flash-frozen in liquid nitrogen and sent to the ESRF (Grenoble, France) for data collection. The dataset was scaled with XDS[42]. Molecular replacement was done using Phaser[43,44]. The structure of native vGalDH was solved using the corresponding AlphaFold2.0-predicted structure as search model. All other structures were solved using the experimental native vGalDH as search model. Structures were refined using COOT[45] and REFMAC5[46] of the CCP4[47] package. The detailed statistics of the crystallographic data are listed in Supplementary Table 3.

## Reporting summary

Further information on research design is available in the Nature Portfolio Reporting Summary linked to this article.

# Data availability

Crystallographic data are deposited with the Protein Data Bank (vGalDH, 8QMY; vGalDH bound to L-galactono-1,4-lactone, 8QNB; A113G vGalDH, 8QNC; G413N vGalDH bound to L-gulono-1,4-lactone, 8QNR). The ancestral sequences of the experimentally characterized enzymes can be found in Supplementary Information and the resulting genes used in this study have been submitted for deposition in the Genbank database (mGULO, PP407300; tGULO, PP407302; jGULO, PP407303; cGULO, PP407301; sGalDH, OQ632496; vGalDH, OQ632497). The collected dataset for the phylogenetic analysis is provided in a Supplementary Data file. All data and materials supporting the findings in the manuscript are available in the Supplementary Information and the Source Data files, and from the corresponding author upon request. Source data are provided with this paper.

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

## Acknowledgements

This research was funded by the ERC Advanced Grant, MetaQ, No. 101094471. M.L.M. was supported by the COFUND project oLife of the European Union's Horizon 2020 research and innovation program under grant agreement No 847675. B-Ligzymes (GA 824017) from the European Union's Horizon 2020 Research and Innovation Program funded A.B. secondment in Groningen, The Netherlands. We acknowledge ESRF for the provision of X- ray beamtime. We thank the staff of the Protein Data Bank for their excellent support during the deposition process.

## Author contributions

M.L.M., M.W.F. and A.M. conceptualized the study and obtained funding. Laboratory work was performed by A.B., N.J., B.M. and M.L.M. Analysis was performed by all authors. The overall supervision for the project was provided by M.W.F. and A.M. A.B. wrote the first draft of the manuscript, and all authors contributed to subsequent revisions and reviewed the final manuscript prior to submission.

## Competing interests

The authors declare no competing interests.
