## [Peer Review File · Nature Communications]

Structure, mechanism, and evolution of the last step in
vitamin C biosynthesisREVIEWER COMMENTS

Reviewer #1 (Remarks to the Author):

Boverio and colleagues present their work that describes the ancestral protein reconstructions of GULO, GalDH and ALO gene families. The authors reveal how substrate specificity (or lack thereof) has evolved between the three families. The authors were also able to crystalize multiple versions of the ancestral GalDH to provide unprecedented details about the structural organization of the substrate binding region and how this region spatially rearranges to accommodate different substrates (super interesting). The results have the potential to be highly valuable to the scientific community, but I have some questions/concerns that need to be addressed before I am fully confident about this potential.

- The authors state that common ancestors were reconstructed with high overall posterior probabilities. However, the PP for sGalDH was only 81%. This is not high. Maybe minimally acceptable. Does this value change when the N-terminal sequences are removed (putative mitochondrial targeting sequence)? Also, the supp mat states that the PP for sGalDH was 93%, not 81%. So there is a mismatch between the article and the supp mat. Please resolve this.

- Cite Randall et al. Nat Commun 2016 7:12847 for the connection between PP and ancient protein phenotypes.

- The authors need to provide a phylogeny with nodal support (posterior probabilities or bootstrap percentages) for the ancestral nodes that are experimentally resurrected in the laboratory. Readers should have confidence in the branching order of the tree. Placement in the supp mat is fine if Figure 1 gets too crowded from this information.

- I perused references 16-19 but it was not obvious to me how sequences from GULO, GalDH and ALO would align to one another. I think it would be helpful if the authors provided an example MSA in the supp mat. Maybe nine total sequences (three diverse representatives from each of the three gene families). This would allow readers to more thoroughly grasp the sequence conversation as opposed to trying to infer it from the branch lengths in Figure 1.

- I assume the authors used the GUI version of PAML. This is not the best version when

other scientists are attempting to replicate someone else's analysis. Thus, it would be helpful if the authors can provide which sequence-model PAML chose as the optimal fit to the data. This information may be listed in one of the output files.

- Supplemental Table 1: Can the authors infer anything about the change in melting temperatures of the ancestral proteins? If not, no need to wildly speculate. I am just wondering, especially as it relates to GULO.

Minor comment:

- Electron appears to be misspelled on P9.

Reviewer #2 (Remarks to the Author):

Boverio et al., comprehensively investigate the origins of substrate specificities for aldonolactone oxidoreductases that catalyze the final oxidative step in the biosynthesis of ascorbic acid. These enzymes are all flavin-dependent oxidoreductases that use sugar substrates and the reductant substrate and molecular oxygen or some other recipient as oxidant substrates. The authors use phylogenic strategies to reconstruct ancestral versions of a number of example enzymes and then characterize the parent, daughter enzymes and specific variant enzymes using steady-state and transient state kinetics augmented with X-ray crystal structures of the holoenzymes, and reductant substrate complexes. The primary achievement is the rather definitive identification of the residues responsible for both reductant and oxidant substrate specificity. The manuscript is logically assembled and, in terms of progression, clearly written. One deficiency is that the manuscript contains no primary kinetic evidence. This may be forgiven as a stylistic matter arising from the nature communication format but this is also true for the supporting information. Despite that the presiding researchers are well-known experts in the study of flavoproteins, it remains important to show evidence of the quality of the observations from which the conclusions arise. This is particularly important for anaerobic work with flavoproteins that frequently accumulate additional phases as a result of imperfect technique and/or added mechanistic complexity.

I offer the following comments and suggested changes:

P2 - 4th line from the bottom of the abstract - "...positions were affixed to specific amino acids..."

P3 - 2nd paragraph

1. "...originated with a few..."

2. "...oxidoreductases. They diverge for stereochemistry...." ("First" is loaded with implications of the order of events)

3. "... (Fig. 1A). They also differ in...." ("Second" has the same problem)

4. "GalDH is a dehydrogenase that reacts poorly with..."

5. "...employs cytochrome c (cyt c) as the oxidant substrate...."

Comment - other than the fact that it can accept electrons from the reduced enzyme, what is the evidence that cytochrome c is a natural oxidant. Low molecular oxygen reactivity is a good indication that some enzymes are dehydrogenases but how can it be known if this is a native interaction? In addition, the source of cytochrome c is not mentioned.

P4 - end of first paragraph - "We reveal that all..."

P5 - halfway down - "We observed only marginal oxidase activity using all...."

P6 - end of first paragraph - "fine geometry" is subjective. It could be a long way from optimal relative to other hydride transfers.

P6 - Second paragraph - why do the oxidases have greater description in the manuscript. For example, why mention the spectral maxima if these were not also described for the dehydrogenases? Similarly, why only conduct transient-state experiments for the oxidases?

P6 - Second paragraph - "Monitoring oxygen consumption,...."

P6 - bottom - ".....mechanism, a pre-steady state kinetic analysis was conducted."

P6 - bottom - "...the rates of the reductive and oxidative half-reactions..."

P7 - top - "The rates of reduction (k_{red}) ranged from $\sim 90 \text{ s}^{-1}$" Describing kinetic events in anything other than finite terms is subjective and not informational.

P7 - middle paragraph "....the fully oxidized state....."

P7 - bottom - "....and electron acceptor..."

P8 - top - "...N5-C4a locus...."

P8 - second paragraph - "....for the reductant substrates...."

P8 - bottom - "....or the respective k_{cat} values...."

P9 - first paragraph - ".....which can no longer form an H-bond..."

P9 - second paragraph - "....residue pair accounts for why..."

P9 - third paragraph - "...substrate cavity made this functional property possible...."

P10 - top - "....enzymes show an early divergence, a dehydrogenase-like lineage (in plants) and an oxidase-like lineage....."

P11 - third paragraph - "...homogenizer (cells were passed three times through the high pressure homogenizer)."

P11 - bottom - ".....they were disrupted by passing through a high-pressure....."

P12 - third paragraph - "Data were processed in...."

P13 - bottom of first paragraph - "....used to remove residual molecular oxygen." ".....were plotted and the resulting dependence was fit to an equation that describes a rectangular hyperbola according to: $k_{obs} = [S]k_{red}/K_s + [S]$ where k_{red} is the limiting rate constant for flavin reduction and K_s is the dissociation constant for the sugar substrate."

Figure 1A - two cytochrome c are reduced per turnover

Figure 2B - the orientation of both structures should be the same

Reviewer #3 (Remarks to the Author):

This is a nice manuscript on the evolutionary origins of vitamin C biosynthetic enzymes. The authors use a variety of approaches, including crystallography, kinetics, mutagenesis, sequence analysis, and ancestral enzyme reconstruction. The work is comprehensive and would be of interest to a wide audience of biochemists. However, there are issues with the crystallography that need to be addressed.

-The validation file for 8QMY shows a close contact between symmetry-related molecules with a distance of 1.9 Å, which is approaching covalent bond distance.

1:B:94:ARG:NH2 1:B:254:GLU:OE1[4_545] 1.90 0.30

The authors should fix this, re-refine, and re-deposit.

-Supplementary Table 3

--add B-factors separately for protein, FAD, and ligand

--add occupancies for FAD and ligand

--The large difference between R (21) and R_{free} (29) 8QNR indicates a potential problem. Please explain/justify.

-Fig. 2 and Fig. 3. The electron density maps should be polder omit maps contoured at 3-4 sigma, not a 2Fo-Fc. See the next comment as well about the maps for the substrates.

-The validation files show very strong (3sigma) negative Fo-Fc density encasing the ligands X8X in 8QNB and X8L in 8QNR. For comparison, for a properly modeled ligand, the density should look like that of the FAD of those validation files. Shockingly, the B-factors for these ligands are zero. Did the B-factors refine to zero? Did they forget to refine the ligand? This is a major flaw and suggests that whoever was responsible for oversight of the person doing the crystallography failed to perform due diligence. This is a major issue because one of the main points of the manuscript assigning roles to specific amino acids, which relies heavily on the enzyme-substrate complex structures. If the ligands in the structures are not valid, the manuscript is undermined.

-It appears that the sugar ligands deposited in the PDB have hydrogen atoms, which is not justified at these resolutions. Note the authors did not include H atoms in the FADs.

-Page 5: The authors could clarify this statement: "These observations show how the side chains lining the catalytic cavity hold the substrate in place to favor the deprotonation of its 2-OH group by the Arg388-Lys456 pair." Do they mean that Arg388 or Lys456 is functioning as a base that deprotonates the 2-OH group? And why does the hydroxyl need to be deprotonated? Are they saying that the 2-OH is deprotonated and reprotonated to give the final product? Perhaps a mechanism diagram would help.

We thank the three Reviewers for their valuable comments. We have incorporated all of them, greatly helping us to further improve the manuscript. Please find below a point-by-point response to the Reviewers' comments in blue and our responses in black.

Reviewer #1

-The authors state that common ancestors were reconstructed with high overall posterior probabilities. However, the PP for sGalDH was only 81%. This is not high. Maybe minimally acceptable. Does this value change when the N-terminal sequences are removed (putative mitochondrial targeting sequence)? Also, the supp mat states that the PP for sGalDH was 93%, not 81%. So there is a mismatch between the article and the supp mat. Please resolve this.

>Thanks for noticing this issue. Indeed, when the deletion of the N-terminus is considered for both plant ancestors the overall PP values improve (around 4%). This information has been added now in the first section of the "Results" as well as in the supplementary figure 2. Also, the mistake in swapping the values of vGalDH and sGalDH posterior probabilities has been corrected. We removed the word "high".

- Cite Randall et al. Nat Commun 2016 7:12847 for the connection between PP and ancient protein phenotypes.

>Done. Reference 26 in the revised manuscript.

- The authors need to provide a phylogeny with nodal support (posterior probabilities or bootstrap percentages) for the ancestral nodes that are experimentally resurrected in the laboratory. Readers should have confidence in the branching order of the tree. Placement in the supp mat is fine if Figure 1 gets too crowded from this information.

>We agree that it is important this information for readers, so the TBE values have been added to Figure 1B at the major divergences in the tree. As the reviewer can see the statistical support is very strong across the entire phylogeny (most of cases TBE= 1).

- I perused references 16-19 but it was not obvious to me how sequences from GULO, GalDH and ALO would align to one another. I think it would be helpful if the authors provided an example MSA in the supp mat. Maybe nine total sequences (three diverse representatives from each of the three gene families). This would allow readers to more thoroughly grasp the sequence conversation as opposed to trying to infer it from the branch lengths in Figure 1.

>As suggested, we have constructed a MSA including three sequences from each type of enzyme (GULOs, ALOs and GalDHs) and added as Supplementary Figure 21. This is referenced in the discussion of the mutations tuning the substrate and electron acceptor specificity.

- I assume the authors used the GUI version of PAML. This is not the best version when other scientists are attempting to replicate someone else's analysis. Thus, it would be helpful if the authors can provide which sequence-model PAML chose as the optimal fit to the data. This information may be listed in one of the output files.

>As indicated in the method section, we only used Paml4.9 and we did not use the GUI version of PAML (PAML-X). The sequence model as well as all other parameters used are specified in the methods section (page 10).

- Supplemental Table 1: Can the authors infer anything about the change in melting

temperatures of the ancestral proteins? If not, no need to wildly speculate. I am just wondering, especially as it relates to GULO.

>The increased thermal stability of some reconstructed sequences is a phenomenon sometimes observed and it does not relate to the protein family *per se*. Instead, it is a bias of the approach, called 'survivor bias' due to the consensus effect. As a reference, we discuss this point in our latest contribution (<https://doi.org/10.1016/j.sbi.2023.102669>). We believe this observation is rather marginal and does not need to be discussed in the context of the manuscript.

-Electron appears to be misspelled on P9.

>Corrected

Reviewer #2

One deficiency is that the manuscript contains no primary kinetic evidence. This may be forgiven as a stylistic matter arising from the nature communication format but this is also true for the supporting information. Despite that the presiding researchers are well-known experts in the study of flavoproteins, it remains important to show evidence of the quality of the observations from which the conclusions arise. This is particularly important for anaerobic work with flavoproteins that frequently accumulate additional phases as a result of imperfect technique and/or added mechanistic complexity.

>As suggested by the Reviewer, the revised Supplementary Figure 19 now includes primary kinetic data for the pre-steady-state kinetics studies on mGULO: the spectral changes during the reductive half-reaction with each tested substrate at 1 mM concentration.

The raw data for kinetics experiments are part of the submission material. The data file also includes examples of the absorbance traces measured with the stopped-flow instrument.

P2 - 4th line from the bottom of the abstract - "...positions were affixed to specific amino acids..."

>Done

P3 - 2nd paragraph

1. "...originated with a few..."

>Done

2. "...oxidoreductases. They diverge for stereochemistry..." ("First" is loaded with implications of the order of events)

>Done

3. "... (Fig. 1A). They also differ in..." ("Second" has the same problem)

>Done

4. "GalDH is a dehydrogenase that reacts poorly with..."

>Done

5. "...employs cytochrome c (cyt c) as the oxidant substrate..."

Comment - other than the fact that it can accept electrons from the reduced enzyme, what is the evidence that cytochrome c is a natural oxidant. Low molecular oxygen reactivity

is a good indication that some enzymes are dehydrogenases but how can it be known if this is a native interaction?

>The revised text states “...is proposed to employ cytochrome c as the natural oxidant substrate because of the physical proximity of GalDH to the respiratory complexes”. See Schertl P, Sunderhaus S, Klodmann J, Grozeff GE, Bartoli CG, Braun HP. L-galactono-1,4-lactone dehydrogenase (GLDH) forms part of three subcomplexes of mitochondrial complex I in *Arabidopsis thaliana*. *J Biol Chem*. 2012;287(18):14412-14419. doi:10.1074/jbc.M111.305144, now cited in the manuscript as reference 22.

In addition, the source of cytochrome c is not mentioned.

>”equine heart” has been added to the relevant text of the Results and Methods sections.

P4 - end of first paragraph - "We reveal that all..."

>Done

P5 - halfway down - "We observed only marginal oxidase activity using all...."

>Done

P6 - end of first paragraph - "fine geometry" is subjective. It could be a long way from optimal relative to other hydride transfers.

>Done: “*This binding modes promotes hydride transfer.....*”

P6 - Second paragraph - why do the oxidases have greater description in the manuscript. For example, why mention the spectral maxima if these were not also described for the dehydrogenases?

>The sentence has been removed.

Similarly, why only conduct transient-state experiments for the oxidases?

>Extant GalDH was already characterized (ref.15) whereas we are not aware of any thorough biochemical and kinetic characterization of GULO published so-far. Moreover, the focus of our work was the oxygen reaction and the evolution of the oxidase activity.

P6 - Second paragraph - "Monitoring oxygen consumption,...."

>Done

P6 - bottom - ".....mechanism, a pre-steady state kinetic analysis was conducted."

>Done

P6 - bottom - "....the rates of the reductive and oxidative half-reactions..."

>Done

P7 - top - "The rates of reduction (k_{red}) ranged from ~90 s⁻¹...." Describing kinetic events in anything other than finite terms is subjective and not informational.

>Done. The exact values and their stand deviations are given.

P7 - middle paragraph "....the fully oxidized state....."

>Done

P7 - bottom - "....and electron acceptor..."

>Done

P8 - top - "...N5-C4a locus...."

>Done

P8 - second paragraph - ".....for the reductant substrates...."

>Done

P8 - bottom - "....or the respective kcat values...."

>Done

P9 - first paragraph- ".....which can no longer form an H-bond..."

>Done

P9 - second paragraph - "....residue pair accounts for why..."

>Done

P9 - third paragraph - "...substrate cavity made this functional property possible...."

>Done

P10 - top - "...enzymes show an early divergence, a dehydrogenase-like lineage (in plants) and an oxidase-like lineage....."

>This sentence was modified as follows: "*an early divergence in two lineages, a dehydrogenase-like one in plants and an oxidase-like one in fungi and metazoa*".

P11 - third paragraph - "...homogenizer (cells were passed three times through the high pressure homogenizer)."

>Done

P11 - bottom - ".....they were disrupted by passing through a high-pressure....."

>Done

P12 - third paragraph - "Data were processed in...."

>Done

P13 - bottom of first paragraph - "....used to remove residual molecular oxygen." ".....were plotted and the resulting dependence was fit to an equation that describes a rectangular hyperbola according to: $k_{obs} = [S]k_{red}/K_s + [S]$ where k_{red} is the limiting rate constant for flavin reduction and K_s is the dissociation constant for the sugar substrate."

>Done

Figure 1A - two cytochrome c are reduced per turnover

>Done

Figure 2B - the orientation of both structures should be the same

>We prefer to keep the two orientations as they are needed for the optimal view of the gating loop conformational change (Figure 2B) and the shape of the active site (Figure 2C).

-The validation file for 8QMY shows a close contact between symmetry-related molecules with a distance of 1.9 Å, which is approaching covalent bond distance.

1:B:94:ARG:NH2 1:B:254:GLU:OE1[4_545] 1.90 0.30

The authors should fix this, re-refine, and re-deposit.

>B:Arg94-B:Glu254 form a tight inter-protein salt bridge that stabilize the crystal packing. We have run an additional round of refinement and the structure was re-deposited. The PDB validation reports are included in the submission.

-Supplementary Table 3

--add B-factors separately for protein, FAD, and ligand

>Done

--add occupancies for FAD and ligand

>Done

--The large difference between R (21) and Rfree (29) 8QNR indicates a potential problem. Please explain/justify.

>The Refmac- and Phenix-calculated Rfree values for the 8QNR structure are 28.3% and 27.7%, respectively. We attribute these slightly higher-than-average values to a higher degree of static disorder in the crystals, possibly induced by the soaking procedure. Indeed, the overall average B-factor is 67 Å². Beside these parameters, the electron densities are of high qualities as demonstrated by the polder map now shown in Figure 3H.

-Fig. 2 and Fig. 3. The electron density maps should be polder omit maps contoured at 3-4 sigma, not a 2Fo-Fc. See the next comment as well about the maps for the substrates.

> The initial manuscript displayed the weighted 2Fo-Fc maps obtained before the inclusion of the ligands in the refinement (e.g. no model bias). Following the Reviewer's advice, the revised manuscript now shows the polder omit maps that were calculated with Phenix (Figures 2A, 3G, and 3H). We notice that the electron density maps are of excellent qualities.

-The validation files show very strong (3sigma) negative Fo-Fc density encasing the ligands X8X in 8QNB and X8L in 8QNR. For comparison, for a properly modeled ligand, the density should look like that of the FAD of those validation files. Shockingly, the B-factors for these ligands are zero. Did the B-factors refine to zero? Did they forget to refine the ligand? This is major flaw and suggests that whoever was responsible for oversight of the person doing the crystallography failed to perform due diligence. This is a major issue because one of the main points of the manuscript assigning roles to specific amino acids, which relies heavily on the enzyme-substrate complex structures. If the ligands in the structures are not valid, the manuscript is undermined.

-It appears that the sugar ligands deposited in the PDB have hydrogen atoms, which is not justified at these resolutions. Note the authors did not include H atoms in the FADs.

>We faced some time-consuming technical issues with the deposition of the two substrate-bound structures. The PDB software did not correctly recognize the ring structure and the chirality of the ligands. Thanks to the wonderful help of the PDB staff (now acknowledged in the revised manuscript) and after various attempts, we first found an interim solution using ligands comprising explicit hydrogen atoms. This allowed us to obtain the validation report needed for initial manuscript submission. Regrettably, we did not add a note in the manuscript submission forms about this point. We apologize for

this omission. While the manuscript was under review, the technical issues were fully solved and the deposited structures were updated to contain the ligand with the refined B-factors and no explicit hydrogens.

“The Materials imply that the same crystallization recipe was used for all four structures, yet the structures reported have at least two (P212121 vs P21) and maybe three (two different P21 lattices) crystal forms. Please confirm the crystallization conditions. Also was a buffer used in the crystallization recipe?”

>The crystals of vGalDH were obtained in conditions 15 (0.2 M lithium nitrate, 20% w/v polyethylene glycol 3,350) and 24 (0.2 M lithium acetate dihydrate, 20% w/v polyethylene glycol 3,350) of the PEG/ion screen (Hampton research). No buffer is present in the reservoir. We changed the text to better specify this point:

“...Elongated needle-like yellow crystals grew in several conditions all containing PEG and lithium salts. The best diffracting crystals were obtained by mixing equal volumes of protein (17 mg vGalDH/ml in 50 mM Hepes pH 7.5) and reservoir (0.2 M LiNO₃ and 20% w/v PEG3350) solutions.....”

Describe the search model used in Phaser and its sequence identity to the targets. Also, was molecular replacement used in all four cases or was Fourier synthesis used in some (i.e., refined model of one structure used as starting point for another structure).”

>the revised “Crystal structure determination” section now states that “....The structure of native vGalDH was solved using the corresponding AlphaFold structure as search model. All other structures were solved using the experimental native vGalDH as search model”

REVIEWER COMMENTS

Reviewer #1 (Remarks to the Author):

The authors have addressed all of my concerns and questions in the revised manuscript.

Reviewer #2 (Remarks to the Author):

Boverio et al., have provided a revised manuscript that has incorporated or responded to all reviewer comments and corrections. Overall I am satisfied with the response save for one detail. Despite piecing through every supporting file provided, I could not locate any raw transient state data as indicated in the response. While an excel file is included that summarizes the findings from transient state experiments (that indicates suitably thorough experimental approach) the time-dependent changes in absorbance traces, as is indicated in the response, are absent. The analysis summary provided suggests rate constants were determined from linear fitting methods which has long ago been superseded by non-linear least squares fitting to combinations exponential terms. In addition, The method of analysis is not detailed in the methods section as the authors skip directly to kobs without explaining how it was measured. Further along these lines, the concentrations of enzyme used in these experiments is not indicated in the methods or the legend for S19 (despite the concentrations of all other components being included, even dioxygen scrubbing enzymes). These are important details to verify that pseudo first order conditions were attained (as the presumed analysis method requires). In my opinion these remain important omissions. I will leave the determination in regard to these issue to the editor. I have no significant additional comment.

Reviewer #3 (Remarks to the Author):

The authors addressed the main issues about the X-ray crystallography; however, new issues have been revealed, which need additional work, and perhaps they forgot to address one of my comments.

-The validation file for 8QMY still shows a problem. Section 5.2 lists a close contact of 1.67 Angstrom between the N and CD atoms of Pro238 in chain C, which is difficult to

understand.

1:C:238:PRO:N 1:C:238:PRO:CD 1.67 1.28

Normally, the clash table lists only the close contacts between nonbonded atoms, yet N and CD of proline are bonded. The table also lists the close contact between C of the preceding residue and CD of Pro238, which may be related (1:C:237:LEU:C 1:C:238:PRO:CD). Perhaps there is a problem with the peptide bond between 237 and 238. The authors need to address this issue, which will require another revision of the deposition.

-Now that the B-factors of the ligands have refined, we see that one ligand is well-resolved by the electron density, while the other is less so. The median B of X8X in 8QNB is quite reasonable at 46, and the density nicely fits the ligand. The authors are justified in writing, "Soaking in a substrate-containing solution led to the bleaching of the crystals and the resulting electron density clearly indicated the presence of the bound substrate". In contrast, the B-factors of X8L in 8QNR range from 78 to 118 with a median of 99, and the density is weak compared to X8X. The authors should include a statement analogous to the one quoted above (perhaps near line 266) stating that the electron density for X8L is not as good and the resulting B-factors are substantially higher, and that their conclusions about the pose of the ligand and its interactions with the protein are less certain. These revisions will improve the transparency of the presentation.

The authors did not address one of my comments from the previous review:

-Page 5: The authors could clarify this statement: "These observations show how the side chains lining the catalytic cavity hold the substrate in place to favor the deprotonation of its 2-OH group by the Arg388-Lys456 pair." Do they mean that Arg388 or Lys456 is functioning as a base that deprotonates the 2-OH group? And why does the hydroxyl need to be deprotonated? Are they saying that the 2-OH is deprotonated and reprotonated to give the final product? Perhaps a mechanism diagram would help.

We thank the three Reviewers for their valuable comments. We have incorporated all of them, greatly helping us to further improve the manuscript. Please find below a point-by-point response to the Reviewers' comments in blue and our responses in black.

Reviewer #1

The authors have addressed all of my concerns and questions in the revised manuscript.

>Many thanks.

Reviewer #2

Boverio et al., have provided a revised manuscript that has incorporated or responded to all reviewer comments and corrections. Overall I am satisfied with the response save for one detail. Despite piecing through every supporting file provided, I could not locate any raw transient state data as indicated in the response. While an excel file is included that summarizes the findings from transient state experiments (that indicates suitably thorough experimental approach) the time-dependent changes in absorbance traces, as is indicated in the response, are absent.

> As suggested by the Reviewer, the raw data for all kinetic traces are now included in the Excel raw data file,

The analysis summary provided suggests rate constants were determined from linear fitting methods which has long ago been superseded by non-linear least squares fitting to combinations exponential terms.

> The Excel raw data file contains examples demonstrating how the data were fitted (see raw data figure S19). As can be seen from these examples, fitting was done using exponential decay functions. This has also been clarified in the Methods section as well.

In addition, the method of analysis is not detailed in the methods section as the authors skip directly to kobs without explaining how it was measured.

>The revised Methods section indicates that the " k_{ox} values were obtained by fitting the transient kinetic traces obtained with the photomultiplier tube detector with standard first- or second-order exponential decay functions using the software linked to the stopped-flow instrument (Applied Photophysics, Surrey, UK)."

Further along these lines, the concentrations of enzyme used in these experiments is not indicated in the methods or the legend for S19 (despite the concentrations of all other components being included, even dioxygen scrubbing enzymes). These are important details to verify that pseudo first order conditions were attained (as the presumed analysis method requires). In my opinion these remain important omissions. I will leave the determination in regard to these issue to the editor. I have no significant additional comment.

> For the stopped-flow measurements, enzyme concentrations were 5-10 μ M after mixing. This is now mentioned in the methods section and in the legends of Figure 3E-F and Supplementary Figure 19. With these conditions, we assured pseudo first order conditions, as the Reviewer rightfully indicates.

Reviewer #3

The authors addressed the main issues about the X-ray crystallography; however, new issues have been revealed, which need additional work, and perhaps they forgot to address one of my comments.

-The validation file for 8QMY still shows a problem. Section 5.2 lists a close contact of 1.67 Angstrom between the N and CD atoms of Pro238 in chain C, which is difficult to understand.

1:C:238:PRO:N 1:C:238:PRO:CD 1.67 1.28

Normally, the clash table lists only the close contacts between nonbonded atoms, yet N and CD of proline are bonded. The table also lists the close contact between C of the preceding residue and CD of Pro238, which may be related (1:C:237:LEU:C 1:C:238:PRO:CD). Perhaps there is a problem with the peptide bond between 237 and 238. The authors need to address this issue, which will require another revision of the deposition.

>Residues 239-247 are disordered in the wild-type crystals. The slightly distorted geometry of Pro238 in subunit C simply reflected the weaker density in this region. Following the Reviewer's request, the issue has been corrected, the coordinates re-deposited in the PDB, and the newly generated validation reported included in the revised manuscript material.

-Now that the B-factors of the ligands have refined, we see that one ligand is well-resolved by the electron density, while the other is less so. The median B of X8X in 8QNB is quite reasonable at 46, and the density nicely fits the ligand. The authors are justified in writing, "Soaking in a substrate-containing solution led to the bleaching of the crystals and the resulting electron density clearly indicated the presence of the bound substrate". In contrast, the B-factors of X8L in 8QNR range from 78 to 118 with a median of 99, and the density is weak compared to X8X. The authors should include a statement analogous to the one quoted above (perhaps near line 266) stating that the electron density for X8L is not as good and the resulting B-factors are substantially higher, and that their conclusions about the pose of the ligand and its interactions with the protein are less certain. These revisions will improve the transparency of the presentation.

>Following the Reviewer's advice, we tempered the tone of the description of substrate binding in 8QNB:

"We further performed the crystallographic analysis of G413N vGalDH at 2.4 Å resolution using crystals that were soaked in a solution comprising L-gulono-1,4-lactone. The B-factors of the bound sugar were notably high (Supplementary Table 3). However, despite this, the electron density was of sufficient quality to conclude that the conformation and orientation of L-gulono-1,4-lactone are highly similar to that of L-galactono-1,4-lactone bound to the wild-type enzyme, with the obvious exception of the 3-hydroxyl group position (Fig. 1A and 3H). The substrate-binding environment is likewise mostly retained in the mutant structure....."

-Page 5: The authors could clarify this statement: "These observations show how the side chains lining the catalytic cavity hold the substrate in place to favor the deprotonation of its 2-OH group by the Arg388-Lys456 pair." Do they mean that Arg388 or Lys456 is functioning as a base that deprotonates the 2-OH group? And why does the hydroxyl need to be deprotonated? Are they saying that the 2-OH is deprotonated and reprotonated to give the final product? Perhaps a mechanism diagram would help.

>The role of Arg388 and homologous residues in the vanillyl-alcohol oxidase family has been analyzed and discussed in reference 31. As suggested by the Reviewer, we have

clarified the text, added a mechanistic diagram (new Figure 2D), and included a new reference about the tautomeric forms of ascorbic acid (ref. 32). We thank the Reviewer because the ascorbic acid tautomerism was not explicitly mentioned in the original manuscript:

“...Arg388 and Lys456 critically position their positively charged side chains in contact with the substrate O2 atom, confirming previous hypotheses about the importance of Arg388 for enzyme activity³¹. In particular, these residues may promote the binding of the alkoxide anion of the substrate, which in turn facilitates hydride transfer from the C2 position to the flavin N5, leading to substrate oxidation (Fig. 2D). The resulting 1,2-diketo product will then spontaneously tautomerize to the more stable enediol form of vitamin C³².

REVIEWERS' COMMENTS

Reviewer #2 (Remarks to the Author):

I thank the authors for their responses to my comments. I am satisfied that the manuscript incorporates edits appropriate to my concerns.

Reviewer #3 (Remarks to the Author):

The authors did a good job of addressing my questions/concerns.